# Schoolteachers and Vaccinations: A Cross-Sectional Study in the Campania Region

**DOI:** 10.3390/vaccines10091519

**Published:** 2022-09-14

**Authors:** Concetta Paola Pelullo, Francesco Corea, Giorgia Della Polla, Francesco Napolitano, Gabriella Di Giuseppe

**Affiliations:** 1Department of Experimental Medicine, University of Campania “Luigi Vanvitelli”, Via Luciano Armanni 5, 80138 Naples, Italy; 2Health Direction, Teaching Hospital, University of Campania “Luigi Vanvitelli”, Via Costantinopoli 104, 80138 Naples, Italy

**Keywords:** attitudes, behaviors, knowledge, school, teacher, vaccination

## Abstract

Background: This cross-sectional survey aimed to determine the knowledge, attitudes, and behaviors regarding vaccinations among schoolteachers in Italy. Methods: Data were collected through an online questionnaire from September 2020 to May 2021 from a sample of schoolteachers in the Campania region of southern Italy. Results: Only 27% of the participants had good knowledge about the vaccinations recommended for schoolteachers. Females who had children, who were unsatisfied by their health status, and not needing additional information about vaccinations were more likely to have good knowledge. Moreover, 61.5% perceived that vaccinations are useful in preventing infectious diseases, and 44.5% agreed or strongly agreed that vaccinations should be mandatory for schoolteachers. The results of multivariate logistic regression showed that schoolteachers who perceived that vaccinations are useful in preventing infectious diseases, who considered the recommended vaccinations to be useful to protecting their own and their students’ health, who believed that information received about vaccination was useful, and who needed additional information about vaccinations recommended for schoolteachers, were significantly more likely to agree or strongly agree that vaccinations should be mandatory for schoolteachers. Conclusions: These findings suggest the need for education strategies to ensure that schoolteachers are better informed about vaccinations recommended for their profession.

## 1. Introduction

Vaccinations represent one of the most effective tools for preventing infectious disease and protecting health [1,2]. In Italy, even before the COVID-19 pandemic there was an increase in the phenomenon of vaccine hesitancy [3,4,5]. The debate about vaccine hesitancy has been much harsher in the last two years, highlighting the factors associated with this phenomenon such as lack of adequate information, concern about the effectiveness and safety of vaccines, and conditioning by the media [6]. Moreover, vaccine hesitancy is a main barrier to vaccination uptake worldwide, and several investigations conducted in Italy have shown vaccine hesitancy ranging from 7.7% to 34.7% [3,7,8,9,10]. Vaccine hesitancy is also associated with sociocultural factors, lack of trust in and information about vaccines, negative attitudes toward the usefulness of vaccinations, and concerns about vaccine side effects, and it may negatively affect coverage in several at-risk population groups, including teachers [11,12,13].

The practice of vaccination, widely shared and implemented in the pediatric sector, is not yet sufficiently widespread in other age groups, such as adolescents, adults, and the elderly, or in some professional categories. In Italy, the National Vaccination Prevention Plan (PNPV) provides recommendations for specific vaccinations for adults and workers at high risk of exposure to infectious diseases [14]. Schoolteachers are one of these at-risk occupational groups due to their close contact with children and adolescents. Indeed, the latest available data on coverage in Italy indicates that, particularly after early childhood and up to adolescence, vaccination coverage is worryingly inadequate [15], increasing the risk of the spread of vaccine-preventable diseases in the school population. Therefore, it is important to explore the knowledge, attitudes, and behaviors surrounding vaccinations among schoolteachers in order to plan interventions to improve knowledge about vaccinations and promote their acceptance in the school population.

Moreover, schoolteachers are an important point of reference for students and families, as they approach them with confidence to ask their opinion on issues of common interest, and teachers support students beyond their school skills [16]. Their role, therefore, could be positive in overcoming any hesitation about vaccinations.

Although several studies have been conducted to evaluate knowledge, attitudes, and behavior towards vaccinations in different settings [17,18,19,20], analysis of the literature has highlighted few experiences [21,22,23] regarding vaccinations recommended for schoolteachers, and the majority have mainly evaluated influenza and COVID-19 vaccinations [12,24,25,26,27,28].

Therefore, the objectives of this study were to investigate knowledge, attitudes, and behaviors about vaccinations among schoolteachers in order to be able to generate insight that may lead to interventions that improve vaccination rates in this professional category.

## 2. Materials and Methods

### 2.1. Study Design and Setting

This study was conducted from September 2020 to May 2021 among a sample of teachers randomly selected from kindergarten, primary, middle, and high schools in the Campania region, in the south of Italy. The sample was selected through a two-stage cluster sampling. In particular, from the list of all eligible schools in the Campania region, eight of them were randomly selected, 6 kindergarten, primary, and middle schools and 2 high schools, for a total of 782 eligible teachers and of a total of about 4000 students. In the second stage, teachers were recruited from each school. Before the initiation of the study, the sample size was calculated, considering a prevalence of 50% of teachers who knew the recommended vaccinations for their professional category, a margin of error of 5%, and a 95% confidence level. Therefore, the minimum total sample size was estimated at 384 schoolteachers.

After school selection, a letter with a request for collaboration was sent to the heads of the selected schools, stating the purpose and the procedure of the survey. After approval, the schoolteachers received an email from the head of the school containing a link to the Google Drive platform providing access to the anonymous online questionnaire. The questionnaire contained a letter explaining that participation in the survey was voluntary and assuring them that privacy and confidentiality were strictly protected. Moreover, it was specified that sending back the questionnaire would be considered an implicit consent to participation. In order to improve the response rate, reminders were sent after two weeks. The participants did not receive any monetary compensation for survey completion.

### 2.2. Survey Instruments

The questionnaire, prepared ad hoc after a review of the literature [18,22,23,29,30], was divided into five sections: (1) sociodemographic and professional characteristics (gender, age, marital status, education level, type of school, years in practice, children, number and characteristics of cohabitants, and perception of personal health status); (2) knowledge of recommended vaccinations for schoolteachers; (3) attitudes toward vaccinations (perceptions of usefulness of vaccinations in preventing infectious diseases, concerns about safety of vaccinations, perception about usefulness of vaccinations for schoolteachers for their own and students’ health, beliefs about mandatory vaccinations for schoolteachers, perception of the risk of transmitting a vaccine-preventable infectious disease to students during their activities, and reliability of the information received about vaccinations); (4) behaviors regarding vaccinations (vaccinations received in the last 10 years); and (5) sources and usefulness of information received and need for additional information about recommended vaccinations for schoolteachers. The overall knowledge score for recommended vaccinations for schoolteachers was determined by assigning 1 point for each correct answer, with a total knowledge score ranging from 0 to 17. Then the median of the total knowledge score was calculated, and respondents with a score above the median were considered to have good knowledge. Perception of the usefulness of vaccinations in preventing infectious diseases was measured on a 10-point Likert scale ranging from 1 (not at all useful) to 10 (extremely useful), and for the analysis this variable was dichotomized as 1–9 (not useful) = 0 and 10 (very useful) = 1. Concerns about the safety of vaccinations were measured on a 10-point Likert scale ranging from 1 (not at all worried) to 10 (extremely worried), and for analysis this was dichotomized as 4–10 (very worried) = 0 and 1–3 (not worried) = 1. Beliefs about vaccinations were measured on a 5-point Likert-type scale with 1 indicating strongly disagree, 2 disagree, 3 uncertain, 4 agree and 5 strongly agree, and for the analysis this variable was dichotomized as strongly disagree/disagree/uncertain = 0 and strongly agree/agree = 1. Finally, self-reported health status was measured on a 10-point Likert scale ranging from 1 (very unsatisfactory) to 10 (very satisfactory), and then dichotomized as satisfactory (10) = 1, and not satisfactory (1–9) = 0.

Before starting the study, to evaluate the correct interpretation, clarity, and reliability of the questions, a pilot study was conducted on 50 teachers.

The study protocol was approved by the Ethics Committee of the University of Campania “Luigi Vanvitelli” (0008667/i 15 April 2020).

### 2.3. Statistical Analysis

All statistical analyses were performed using Stata statistical software version 15 [31]. First, the results of the descriptive analysis were reported as frequency percentages and means. Second, bivariate appropriate tests (*t*-test and chi-square test) were conducted to assess the associations between each of the independent characteristics and the outcomes of interest. Then, only variables found to be associated at a *p*-value ≤ 0.25 were introduced into multivariate logistic regression models [32]. The stepwise procedure was set at a *p*-value for entry and exclusion of 0.2 and 0.4, respectively. Odds ratios (ORs) and 95% confidence intervals (CIs) were presented in all logistic and ordered regression models.

Multivariate regression analysis was conducted to examine which of the characteristics were significantly associated with the following main outcomes: (1) good knowledge of recommended vaccinations for schoolteachers (no = 0; yes = 1) (Model 1); (2) perception of the usefulness of vaccinations in preventing infectious diseases (ordinal) (Model 2); (3) belief that vaccinations should be mandatory for schoolteachers (ordinal) (Model 3); and (4) having been informed by their physicians about recommended vaccinations for their professional category (no = 0; yes = 1) (Model 4). In all the models, the following independent variables were included: gender (male = 0; female = 1); age in years (23–44 = 1; 45–55 = 2; >55 = 3); marital status (unmarried/separated/divorced/widowed = 0; married/cohabitant = 1); education level (high school = 0; university degree/master’s degree = 1); type of school (kindergarten/primary = 1; middle school = 2; high school = 3); number of years in practice (1–15 = 1; 16–29 = 2; ≥30 = 3); number of cohabitants (continuous); having children (no = 0; yes = 1); family members under the age of 12 years (no = 0; yes = 1); family members over the age of 64 years (no = 0; yes = 1); relatives with underlying chronic medical conditions (no = 0; yes = 1); having a satisfactory self-reported health status (no = 0; yes = 1); physicians as sources of information about vaccinations (no = 0; yes = 1); usefulness of information received about vaccinations (no = 0; yes = 1); and need for additional information (no = 0; yes = 1). Moreover, the following variables were also included: good knowledge of vaccinations recommended for schoolteachers (no = 0; yes = 1) in Models 2 and 3; perception of usefulness of vaccinations in preventing infectious diseases (no = 0; yes = 1); perception of the usefulness of vaccinations for schoolteachers for their own health (no = 0; yes = 1); and beliefs about the usefulness of vaccinations for schoolteachers for their students’ health (no = 0; yes = 1) in Model 3.

## 3. Results

### 3.1. Sociodemographic and Professional Characteristics of the Study Population

Of the 782 schoolteachers invited, 519 agreed to participate in the survey, with a response rate of 66.4%. Table 1 shows the main sociodemographic characteristics of the sample. The majority were women (88.8%), the average age was 51.5 years (range = 23–67), and 76.7% were married/cohabitant. In regard to professional characteristics, the majority (39.1%) worked in middle schools, and the average number of years in practice was 19.4 (range = 1–43). Moreover, 76.7% of respondents had more than one cohabitant, 79% had children, 25.2% had relatives with underlying chronic medical conditions, and 22% and 24.2% had family members under the age of 12 years and over the age of 64 years, respectively. Finally, only 11.8% of schoolteachers were satisfied with their health status, with an overall mean value of 7.8 out of a maximum value of 10.

### 3.2. Knowledge of Vaccinations Recommended for Schoolteachers

When knowledge of recommended vaccinations for schoolteachers was investigated, the correct responses ranged from 21% for influenza to 48.8% for tetanus vaccinations. Overall, only 27% of the respondents had good knowledge of recommended vaccinations for schoolteachers, and the results of the multiple logistic regression analyses showed that this knowledge was significantly higher in females (OR = 3.01; 95% CI 1.24–7.32), who had children (OR = 2.1; 95% CI 1.17–3.76), who were unsatisfied by their self-reported health status (OR = 0.44; 95% CI 0.2–0.98), and who did not need additional information about vaccinations (OR = 0.47; 95% CI 0.29–0.79) (Model 1 Table 2).

### 3.3. Attitudes towards Vaccinations

In regard to attitudes, 41.4% of respondents had no concerns about the safety of vaccinations, with a median value of 4, and 61.5% perceived that vaccinations are useful in preventing infectious diseases. The results of the multivariate logistic regression model showed that having fewer cohabitants (OR = 0.8; 95% CI 0.66–0.98), having a family member under the age of 12 years (OR = 1.89; 95% CI 1.11–3.21), having good knowledge of recommended vaccinations for schoolteachers (OR = 1.91; 95% CI 1.21–3.03), and believing that information received about vaccinations was useful (OR = 2.88; 95% CI 1.76–4.70) were significant predictors of the perception that vaccinations are useful in preventing infectious diseases. (Model 2 Table 2).

More than half of the teachers (58.6%) were in favor of vaccinations and considered the recommended vaccinations useful in protecting their (61.8%) and their students’ health (62.4%). Moreover, 76.5% considered themselves at high risk of transmitting an infectious disease to the students during teaching activities, 60.9% believed that there is a need to improve confidence in vaccinations, and 42.8% agreed or strongly agreed that the information received about vaccinations was reliable.

Moreover, 44.5% agreed or strongly agreed that vaccinations should be mandatory for schoolteachers. The results of the multivariate logistic regression showed that those who believed that vaccinations are useful in preventing infectious diseases (OR = 1.84; 95% CI 1.28–2.64), who considered the recommended vaccinations useful in protecting their (OR = 4.35; 95% CI 1.82–10.4) and their students’ health (OR = 2.84; 95% CI 1.20–6.71), and who believed that information received about vaccinations was useful (OR = 1.59; 95% CI 1.07–2.38) were significantly more likely to agree or strongly agree that vaccinations should be mandatory for schoolteachers. Furthermore, respondents aged 23-44 years (OR = 0.54; 95% CI 0.34–0.87) and 45–55 years (OR = 0.61; 95% CI 0.40–0.91) were less likely to have this positive attitude compared to those aged more than 55 years (Model 3 Table 2).

### 3.4. Behavior towards Vaccination

Only 23.7% were informed by their physicians about recommended vaccinations for their professional category. The results of the multivariate logistic regression showed that those who had relatives with underlying chronic medical conditions (OR = 1.7; 95% CI 1.04–2.79), and who had received information from physicians about vaccinations (OR = 2.8; 95% CI 1.71–4.59) were significantly more likely to have been informed by their physicians about vaccinations recommended for them. Moreover, those who worked at high schools (OR = 0.47; 95% CI 0.26–0.85) were less informed by their physicians about recommended vaccinations for their professional category compared to those who worked at kindergarten/primary schools (Model 4 Table 2). When asked to indicate which vaccine-preventable infectious diseases they had contracted during their life, the participants indicated, in descending order: measles (79.2%), varicella (78.2%), rubella (58.8%), pertussis (40.5%), mumps (53.4%), and hepatitis B (3.1%). Moreover, 31% of respondents reported having received at least one vaccination for influenza, and 11.4% for diphtheria, tetanus, and pertussis in the last 10 years.

### 3.5. Sources of Information

Overall, 98.6% of schoolteachers had received information about vaccination: 59.9% from physicians, 51.6% from the Internet, 49.3% from TV, and 15.2% from social networks. Only 31.2% of respondents had been informed by their physicians about influenza vaccination. Moreover, 24.8% of respondents judged the information received as useful, and 81.6% expressed the need for additional information about recommended vaccinations for schoolteachers.

## 4. Discussion

To the best of our knowledge, this study is one of the first conducted to specifically investigate the knowledge, attitudes, and behavior toward recommended vaccinations for schoolteachers in Italy. Comparison with other studies is difficult because of available samples and different methodologies used.

Although the Italian PNPV recommends vaccinations for different professional categories, such as schoolteachers, our research findings revealed inaccurate knowledge about these vaccinations. Indeed, the study showed that only 27% of schoolteachers knew the vaccinations recommended for them. In comparison with similar previous studies conducted in different countries, this rate was lower than that reported in the USA in a survey among schoolteachers, in which a total of 40% gave incorrect answers about such knowledge [24]. By contrast, a considerably higher value has been observed in two surveys conducted in the USA regarding influenza vaccination [21] and in Turkey about HPV [33]. Moreover, although in different populations, a higher level of knowledge was found among Italian adolescents, with 57.2% of respondents having a fair/satisfactory knowledge of vaccine-preventable diseases [18], and among parents, with 31.4% of the respondents who had good knowledge of the main mandatory and recommended vaccinations for 6 month–6 year-old children [34]. This low level of knowledge is rather worrying because the indications contained in the PNPV are clear and were approved several years ago. This shows that there has been a lack of proper dissemination of information in this professional category.

Overall, attitudes toward vaccinations were generally positive. Indeed, more than half were favorable toward vaccinations and considered the recommended immunizations for schoolteachers useful in protecting their own and their students’ health. These positive attitudes, joined with the perception of the risk of transmitting an infectious disease during their activities with their students, could be a facilitator in approaching vaccinations in a conscious way. Moreover, multivariate analysis has made it possible to identify positive predictors related to attitudes, highlighting that the information received, in particular that deemed to be of quality, has positively influenced attitudes about the usefulness of vaccinations and making them mandatory. When we investigated this attitude, 44.5% expressed their agreement with this statement. This finding was slightly lower than what is reported in two studies conducted in the USA, in which 45.7% and 46.3% believed that vaccinations should be mandatory [22,23]. The approach to the protection of one’s health has been changing over time, going from a predominantly passive behavior to a decidedly active attitude, in which the subject decides what to do for his/her own health and consciously accepts or rejects the proposals of physicians. This has also occurred in the debate over vaccinations, moving from a passive acceptance of mandatory vaccinations to a questioning of their mandatory nature to an informed and conscious choice or refusal. Therefore, it is necessary to increase confidence and awareness regarding vaccinations, primarily among populations at high risk of contracting vaccine-preventable infectious diseases.

Moreover, only 31% of respondents reported having received at least one vaccination for influenza, and 11.4% for diphtheria, tetanus, and pertussis in the last 10 years. In particular, influenza vaccination coverage is alarmingly low, considering everyday contact with students and the subsequent possibility of spreading influenza within the context of schools. Influenza vaccination uptake was lower compared to that observed in other populations [29,35] but higher than that reported among schoolteachers [25,36]. Moreover, a higher influenza vaccination uptake was found in a study conducted in Greece among schoolteachers, in which 34.8% had received the annual influenza vaccine [26]. Furthermore, influenza vaccination coverage was lower compared to the rate reported among healthcare workers (HCWs). One explanation is that HCWs consider themselves at high risk of contracting influenza and COVID-19 [37,38,39]. This has led over time to the implementation of information campaigns about influenza vaccination in HCWs, also promoting on-site vaccinations in order to improve the uptake [40]. Another main finding from this study is that regarding information received by physicians. This is not surprising because physicians are recognized as a reliable source of information about health issues and represent one of the most important determinants positively influencing awareness of vaccinations in several settings [41,42].

### Limitations

This study has some limitations that need to be considered in interpreting the results. First, the design of the study was based on cross-sectional data, and therefore it is not possible to establish the direction of the associations and temporal relationships between the investigated outcomes and the independent variables. Second, only teachers with an e-mail account were included, but this bias is not relevant because in Italy communications with teachers are no longer by paper, and therefore all teachers are required to have a personal e-mail account. Third, data were collected during the COVID-19 pandemic, particularly in some periods in which in-person learning activity had been suspended, and when teachers’ interest was focused on the mandatory nature of the COVID-19 vaccine, and therefore their attitudes toward vaccination might change with the resumption of normal working activities. Fourth, a recall bias may have occurred, since we investigated vaccination uptake in the last 10 years. These limitations have been contained because the questionnaire was self-administered and therefore had the guarantee of the confidentiality and anonymity of responses. Fifth, the teachers were selected only in one region of Italy, and this can affect the generalizability of the findings. However, we selected only public schools, which represent the vast majority of Italian schools and whose organization is similar throughout the nation. Finally, the sample was selected from different types of schools. Thus, the specific context of work could influence the generalizability of the findings and, therefore, these results might not reflect the knowledge, attitudes, and behaviors of all teachers. Despite these limitations, this survey provides important insights on teachers’ knowledge, attitudes, and behavior toward vaccinations recommended for them.

## 5. Conclusions

In conclusion, this study revealed an inaccurate level of knowledge but general positive attitudes toward vaccinations. Therefore, these findings suggest that communication and education strategies should be undertaken as part of a targeted vaccination program for schoolteachers, particularly during the current COVID-19 pandemic, to improve their knowledge and their compliance with vaccinations, and indirectly that of their students, given the social role of this professional category. Furthermore, a teacher trained in vaccination topics could be encouraged to take an active role in disseminating adequate information about vaccination.

## Figures and Tables

**Table 1 vaccines-10-01519-t001:** Sociodemographic and professional characteristics of the study population.

Characteristics	N	%
**Gender**		
Male	58	11.2
Female	460	88.8
**Age in years**	51.5 ± 9.4 (23–67) *	
23–44	123	23.7
45–55	191	36.8
>55	205	39.5
**Marital status**		
Unmarried/separated/divorced/widowed	120	23.3
Married/cohabitant	394	76.7
**Education level**		
High school	117	22.8
University degree/master’s degree	397	77.2
**Type of school**		
Kindergarten/primary school	197	38
Middle school	203	39.1
High school	119	22.9
**Number of years in practice**	19.4 ± 10.9 (1–43) *	
1–15	197	38.5
16–29	192	37.5
≥30	123	24
**Number of cohabitants**		
0	21	4.1
1	99	19.2
2	157	30.5
3	186	36.1
4	48	9.3
5	4	0.8
**Having children**		
No	108	21
Yes	407	79
**Family members under the age of 12 years**		
No	394	78
Yes	111	22
**Family members over the age of 64 years**		
No	383	75.8
Yes	122	24.2
**Relatives with underlying chronic medical conditions**		
No	378	74.8
Yes	127	25.2
**Having a satisfactory self-reported health status**		
No	456	88.2
Yes	61	11.8

Numbers for each item may not add up to the total number of the study population due to missing values. * Mean ± standard deviation (range).

**Table 2 vaccines-10-01519-t002:** Multivariate logistic regression analyses to characterize factors associated with the outcomes of interest.

**Model 1: Good Knowledge of Recommended Vaccinations for Schoolteachers**	**OR**	**95% CI**	***p*-Value**
Log likelihood = −270.97; χ^2^ = 32.36 (6 *df*); *p* < 0.0001			
Having children			
No	1.00 *		
Yes	2.1	1.17–3.76	0.013
Gender			
Male	1.00 *		
Female	3.01	1.24–7.32	0.015
Having a satisfactory self-reported health status			
No	1.00 *		
Yes	0.44	0.2–0.98	0.044
Need for additional information			
No	1.00 *		
Yes	0.47	0.29–0.79	0.004
Usefulness of information received about vaccination			
No	1.00 *		
Yes	0.7	0.42–1.14	0.152
Type of school			
Kindergarten/primary school	1.00 *		
Middle school	Backward elimination		
High school	0.7	0.41–1.18	0.181
**Model 2. Perception of the Usefulness of Vaccinations in Preventing Infectious Diseases**	**OR**	**95% CI**	***p*-Value**
Log likelihood = −519.82; χ^2^ = 52.91 (12 *df*); *p* < 0.0001			
Number of cohabitants (continuous)	0.8	0.66–0.98	0.028
Family members under the age of 12 years			
No	1.00 *		
Yes	1.89	1.11–3.21	0.019
Age in years			
23–44	0.86	0.43–1.75	0.687
45–55	0.71	0.42–1.18	0.189
>55	1.00 *		
Good knowledge of recommended vaccinations for schoolteachers			
No	1.00 *		
Yes	1.91	1.21–3.03	0.005
Usefulness of information received about vaccination			
No	1.00 *		
Yes	2.88	1.76–4.70	<0.001
Having a satisfactory self-reported health status			
No	1.00 *		
Yes	1.64	0.83–3.24	0.150
Gender			
Male	1.00 *		
Female	1.61	0.91–2.84	0.102
Number of years in practice			
1–15	0.73	0.37–1.42	0.350
16–29	0.9	0.50–1.60	0.712
≥30	1.00 *		
Family members over the age of 64 years			
No	1.00 *		
Yes	1.32	0.83–2.12	0.241
Need for additional information			
No	1.00 *		
Yes	0.72	0.41–1.25	0.210
**Model 3. Belief that Vaccinations Should Be Mandatory for Teachers**	**OR**	**95% CI**	***p*-Value**
Log likelihood = −637.33; χ^2^ = 205.39 (13 *df*); *p* < 0.0001			
Age in years			
23–44	0.54	0.34–0.87	0.010
45–55	0.61	0.40–0.91	0.015
>55	1.00 *		
Education level			
High school	1.00 *		
University degree/master’s degree	1.26	0.85–1.89	0.247
Usefulness of information received about vaccination			
No	1.00 *		
Yes	1.59	1.07–2.38	0.012
Perception of the usefulness of vaccinations in preventing infectious diseases			
No	1.00 *		
Yes	1.84	1.28–2.64	0.001
Perception of the usefulness of vaccinations for schoolteachers for their own health			
No	1.00 *		
Yes	4.35	1.82–10.4	0.001
Perception of the usefulness of vaccinations for schoolteachers for their students’ health			
No	1.00 *		
Yes	2.84	1.20–6.71	0.017
Need for additional information			
No	1.00 *		
Yes	1.45	0.93–2.26	0.103
Good knowledge of recommended vaccinations for schoolteachers			
No	1.00 *		
Yes	0.93	0.53–1.37	0.725
Having a satisfactory self-reported health status			
No	1.00 *		
Yes	0.83	0.48–1.48	0.528
Relatives with underlying chronic medical conditions			
No	1.00 *		
Yes	1.10	0.72–1.69	0.655
Family members over the age of 64 years			
No	1.00 *		
Yes	1.22	0.77–1.92	0.152
Number of cohabitants (continuous)	0.93	0.79–1.10	0.410
**Model 4. Having Been Informed by Their Physicians about Recommended Vaccinations for Their Professional Category**	**OR**	**95% CI**	***p*-Value**
Log likelihood = −247.34; χ^2^ = 33.95 (6 *df*); *p* < 0.0001			
Type of school			
Kindergarten/primary school	1.00 *		
Middle school	Backward elimination		
High school	0.47	0.26–0.85	0.012
Relatives with underlying chronic medical conditions			
No	1.00 *		
Yes	1.7	1.04–2.79	0.034
Physicians as sources of information about vaccinations			
No	1.00 *		
Yes	2.8	1.71–4.59	<0.001
Having children			
No	1.00 *		
Yes	1.55	0.85–2.84	0.157
Usefulness of information received about vaccination			
No	1.00 *		
Yes	1.43	0.88–2.32	0.151
Family members under the age of 12 years			
No	1.00 *		
Yes	1.27	0.75–2.15	0.372

* Reference category.

## Data Availability

The data presented in this study are available on request from the corresponding author.

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
