# Peer review of "Schoolteachers and Vaccinations: A Cross-Sectional Study in the Campania Region"

_vaccines, 2022, doi:10.3390/vaccines10091519_

Round 1

Reviewer 1 Report

The nine months’ cross-sectional survey conducted by Pelullo et al. on the school teachers of Campania region, southern Italy revealed 27% positive response about vaccination which apparently seems to be ambiguous since the Italian Government took a lot of preventive measures with legislative actions against SARS-CoV-2 infection. The teaching community must be well aware of such remedies, and this very work stated a very low response from a substantially educated community. Although more than 60% of teachers had a confidence on vaccine action to prevent COVID-19 pandemic; however, approximately 45% of the total teachers voted for vaccinations. Such a scenario is not acceptable in modern world where scientists are continually giving efforts on the development of new drugs/ vaccines using various platforms to treat the ongoing and upcoming diseases. Such data is useful to ponder to the truth regarding the belief on vaccination possessed by the educated community (in this case the school teachers only). Teachers play a great role on convincing their students on what to do and what not. But when the teachers are found with misconception and low confidence about vaccination, that seems to be dangerous for their students. Italian Government should take stringent steps to convince the teachers about the beneficiary sides of vaccination. Considering such a scenario, I appreciate the work done by Pelullo and colleagues. Certainly a sound work on such survey will be helpful for other countries too. However, some specific points need better clarifications for the sake of the improvement of the manuscript.

1.       In the first paragraph of the Introduction, authors should elaborate more on the hesitancy about vaccination in Italy on a time line basis. The evolution resistance against vaccination, the specific points against vaccination, the corresponding community, and the lack of proper information (if any) on vaccination needs to be described.

2.       Page 2: lines 46-59: It is still unclear to me why the authors chose the school teachers, why not the college teachers or the University Professors. I know authors have a clear objective of inclusion of such participants; but they should write that clearly. An inclusion criteria and the exclusion criteria would be better. Authors should mention those points otherwise the rationale of the work would be lacking.

3.       In Materials and Methods, authors should mention the regions of Italy, the total school number, the number of teachers and students. Choosing a particular state is not good actually. But the authors should justify this point.

4.       In many cases the Google forms are confusing to many people; however, authors should consider a statistical median to evaluate the responses. Such a flexible consideration may even increase the ratio of the positive response.

5.       It would be better if the authors could continue the survey having more teachers since the present sample size is not sufficient. At least 3000 teachers should be provided with the questionnaire; and authors should reconsider the responses. This is not that difficult task; authors shall have to send the google form to more teachers as many as they can.

6.       Lines 206-208: “More than half (58.6%) were favorable ………………………………. during their activity to their students,…………….” Sentences are contradictory. I feel that authors are not clear what they stated here. Please clarify your statement.

7.       Line 223: This is not the fault of the school teacher only; indeed, there should be a system of case history. Authors should consider that point too.

Author Response

Reviewer 1

  1. In the first paragraph of the Introduction, authors should elaborate more on the hesitancy about vaccination in Italy on a time line basis. The evolution resistance against vaccination, the specific points against vaccination, the corresponding community, and the lack of proper information (if any) on vaccination needs to be described.

As suggested, in the Introduction section we have included a paragraph on the hesitancy about vaccination in Italy, the factors associated and its consequences.

  1. Page 2: lines 46-59: It is still unclear to me why the authors chose the school teachers, why not the college teachers or the University Professors. I know authors have a clear objective of inclusion of such participants; but they should write that clearly. An inclusion criteria and the exclusion criteria would be better. Authors should mention those points otherwise the rationale of the work would be lacking.

As suggested, in the introduction section we have included a paragraph in order to explain the rationale of the our study.

  1. In Materials and Methods, authors should mention the regions of Italy, the total school number, the number of teachers and students. Choosing a particular state is not good actually. But the authors should justify this point.

As suggested, in the Materials and Methods section we have included the total school number, and the number of teachers and students of the selected schools. Moreover, in the Limitations section we have discuss the point that the teachers were selected only in one region of Italy.

  1. In many cases the Google forms are confusing to many people; however, authors should consider a statistical median to evaluate the responses. Such a flexible consideration may even increase the ratio of the positive response.

As suggested, we calculated the statistical medians of the values of the continuous variables.

  1. It would be better if the authors could continue the survey having more teachers since the present sample size is not sufficient. At least 3000 teachers should be provided with the questionnaire; and authors should reconsider the responses. This is not that difficult task; authors shall have to send the google form to more teachers as many as they can.

Thank you very much for the suggestion for future surveys. We are aware that the study sample must be representative of the population to warrant accurate generalization of the findings and that a greater number of the study population may increase the power of the study. However, in this survey sample size was calculated accurately, the results showed a good response rate, and therefore, we are confident that study findings are valid.

  1. Lines 206-208: “More than half (58.6%) were favorable ………………………………. during their activity to their students,…………….” Sentences are contradictory. I feel that authors are not clear what they stated here. Please clarify your statement.

As suggested, we rewrote the paragraph for a better understanding of the results.

  1. Line 223: This is not the fault of the school teacher only; indeed, there should be a system of case history. Authors should consider that point too.

As suggested, to avoid misunderstanding, we have moved the results about influenza vaccination information in Source of information section.

Reviewer 2 Report

I’ve read the paper by Pellullo and colleagues, titled “School teachers and vaccinations: a cross-sectional study in Italy”. The topic’s relevant and the study aimed at unrevealing some important predictors of teachers’ KAP towards vaccines and vaccination that can be addressed future health eduction initiatives, and public health policies. 

However, the article requires more deep investigation before considering it for publication. Also, the whole manuscript should undergo thorough English revisions to improve clarity of the message and grammatical structure of the text.

Please, find here a number of suggestions/comments to improve the paper. 

The title reads  “a cross-sectional study in Italy”, but it soon becomes clear from the abstract that the study was limited to an area of Campania region, thus it should be changed accordingly. 

That towards usefulness of vaccination is more a "perception” than a “belief”. Methods and results should consider such a relevant difference. 

For the purpose of their analysis, authors decided to dichotomize most of the 17 variables collected on a Likert scale. I understand the rational behind the strategy, although being an arguable point: it is not clear how they fixed the cut-off for dichotomization? It was different for each variables. In some cases, it has been also considered in a way that flatten the information that could come from the rest of the distribution (being either normal or not): i.g.,  self-reported (lines 112-115), which compared the 11% the participants with the rest of the sample that include 9 different levels of “perceptions”.

In this sense, also the outcome “accurate knowledge” does reflect this differences and cannot be considered “accurate” whatsoever. Authors could change it into “good knowledge”.

Again, another option (just a suggestion) to avoid dichotomization is to use regression analysis for ordinal outcomes (STATA offers a number of options that might suit data from this study). Indeed, Likert scales is really more of an ordinal measure than a continuous measure and therefore doesn’t fit the assumptions of linear regression.

There is a spin bias between text and tables. For instance, in Methods (line 88) authors report to have collected gender, but in table 1, they listed the sex. Sex' and 'gender' are often used interchangeably, but they have different meanings. 

Line 292: I don’t understand the comparison with HCWs: teachers and HCWs are clearly two groups  at increased risk for influenza infection and health and social consequences, and both tend to report a low rate of adherence for vaccine uptake, but how these two groups can be correlated?

The third study acknowledged by the same authors “ Third, data were collected during COVID-19 pandemic, particularly in some periods in which the didactic activity in presence had been suspended, and when teachers’ interest focused on mandatory of COVID-19 vaccine, and, therefore, their attitudes towards vaccination might change to the resumption of normal working activities” opens to possible further investigation. When the global massive vaccination campaigns started, teachers were prioritized for vaccination against COVID-19. However, in this study teachers were enrolled at the turn of the absence of vaccine and the starting of the first administration. Now, also considering the timing of COVID-19 vaccination in their country, authors might want to investigate possible differences in their cohort in the group of teachers enrolled before the authorization of COVID-19 vaccines and in that enrolled after. 

Author Response

Reviewer 2

The title reads  “a cross-sectional study in Italy”, but it soon becomes clear from the abstract that the study was limited to an area of Campania region, thus it should be changed accordingly.

As suggested, we have specified in the title that the survey refers to southern Italy and we have modified the title from “School teachers and vaccinations: a cross-sectional study in Italy” in “School teachers and vaccinations: a cross-sectional study in Campania region”.

That towards usefulness of vaccination is more a “perception” than a “belief”. Methods and results should consider such a relevant difference.

As suggested, we have made the required changes.

For the purpose of their analysis, authors decided to dichotomize most of the 17 variables collected on a Likert scale. I understand the rational behind the strategy, although being an arguable point: it is not clear how they fixed the cut-off for dichotomization? It was different for each variables. In some cases, it has been also considered in a way that flatten the information that could come from the rest of the distribution (being either normal or not): i.g.,  self-reported (lines 112-115), which compared the 11% the participants with the rest of the sample that include 9 different levels of “perceptions”.

Before dichotomization, we have performed the Shapiro-Wilk test, to check if a continuous variable follows a normal distribution. Since the null hypothesis was rejected, we dichotomized the variables.

In this sense, also the outcome “accurate knowledge” does reflect this differences and cannot be considered “accurate” whatsoever. Authors could change it into “good knowledge”.

As suggested, we have made the required changes.

Again, another option (just a suggestion) to avoid dichotomization is to use regression analysis for ordinal outcomes (STATA offers a number of options that might suit data from this study). Indeed, Likert scales is really more of an ordinal measure than a continuous measure and therefore doesn’t fit the assumptions of linear regression.

As suggested, we have performed ordered logistic analysis for ordinal outcomes.

There is a spin bias between text and tables. For instance, in Methods (line 88) authors report to have collected gender, but in table 1, they listed the sex. Sex' and 'gender' are often used interchangeably, but they have different meanings.

As suggested, we have made the required changes.

Line 292: I don’t understand the comparison with HCWs: teachers and HCWs are clearly two groups at increased risk for influenza infection and health and social consequences, and both tend to report a low rate of adherence for vaccine uptake, but how these two groups can be correlated?

Our study is one of the few studies conducted on vaccinations and teachers, so it was difficult to make comparisons within the same population. For this reason, our results were compared with HCWs, which are another risk group for influenza infection and its consequences.

The third study acknowledged by the same authors “ Third, data were collected during COVID-19 pandemic, particularly in some periods in which the didactic activity in presence had been suspended, and when teachers’ interest focused on mandatory of COVID-19 vaccine, and, therefore, their attitudes towards vaccination might change to the resumption of normal working activities” opens to possible further investigation. When the global massive vaccination campaigns started, teachers were prioritized for vaccination against COVID-19. However, in this study teachers were enrolled at the turn of the absence of vaccine and the starting of the first administration. Now, also considering the timing of COVID-19 vaccination in their country, authors might want to investigate possible differences in their cohort in the group of teachers enrolled before the authorization of COVID-19 vaccines and in that enrolled after. 

This study has been conducted from September 2020 to May 2021. Covid-19 vaccine was available for teachers from 9 April 2021 and it became mandatory from 12 December 2021 for this population. At the time of the survey the vaccine strategic plan did not include the teachers completely and for this population the vaccination offer was in a very early stage.

Round 2

Reviewer 1 Report

Authors have addressed my queries properly.

Reviewer 2 Report

I have read the new version of the ms, now titled “School teachers and vaccinations: a cross-sectional study in Campania region”.

I really appreciate authors’ efforts in improving their work. There is nonetheless an issue regarding the dichotomization. Authors replied “Before dichotomization, we have performed the Shapiro-Wilk test, to check if a continuous variable follows a normal distribution. Since the null hypothesis was rejected, we dichotomized the variables.”, but this does not answer to my previous comments. Anyway, if the Editor considers this response satisfactory, for me it is ok.